# Association between low muscle mass and metabolic syndrome in elderly Japanese women

**Kazushi Nomura**[1]*, **Masato Eto**[1], **Sumito Ogawa**[1], **Taro Kojima**[1], **Katsuya Iijima**[1], **Tetsuro Nakamura**[2], **Atsushi Araki**[3], **Yasuyoshi Ouchi**[1,4], **Masahiro Akishita**[1]

1 Department of Geriatric Medicine, The University of Tokyo Graduate School of Medicine Faculty of Medicine, Tokyo, Japan, 2 Research Institute of Aging Science, Abe Clinic, Tokyo, Japan, 3 Department of Endocrinology, Tokyo Metropolitan Geriatric Hospital, Tokyo, Japan, 4 Toranomon Hospital, Tokyo, Japan

* k-nomura@umin.ac.jp

**Data Availability Statement:** All relevant data are within the manuscript and its Supporting Information files.

## Abstract

Metabolic syndrome (MetS) is an important predictor of mortality in older adulthood, but it is not reliably related to measures of body composition such as body mass index in older adults, as opposed to those in earlier life stages. Previous research suggests that skeletal muscle mass is related to cardiovascular risk in older adulthood, but it is difficult to measure muscle mass accurately and independently of body fat. This study aimed to examine the relationship between body composition and cardiovascular risk factors among women in older adulthood. A cross-sectional observational clinical study was conducted at a single medical clinic in Tokyo, Japan. Participants included 90 healthy Japanese women aged 65 years and older. MetS risk factors were assessed. Appendicular skeletal muscle mass (ASM) was assessed using dual-emission X-ray absorptiometry. Visceral fat area (VFA) was measured using computed tomography. VFA positively correlated with ASM and MetS, whereas ASM and MetS did not correlate with each other. Using VFA and ASM data in a MetS multiple linear regression model, the association between VFA and MetS remained positive, whereas a significant negative relationship emerged between ASM and MetS. Lower muscle mass was independently associated with higher cardiovascular risk after controlling for VFA. Clinical interventions to reduce muscle loss in older adulthood may be beneficial for reducing the risk of MetS and improving cardiovascular health.

## Introduction

Metabolic syndrome (MetS) is a common disorder related to insulin resistance and obesity, which has significant impacts on a number of negative health outcomes via inflammatory pathways [1]. In Japan, MetS is an especially serious problem among older women, with a prevalence of 20.7%, and has been cited by the Japanese Ministry of Health as a major cause of rising health care costs [2]. MetS has been found to contribute to serious health problems related to aging, including cardiovascular disease [3] and dementia [4]. More than 27% of the Japanese population is over the age of 65, with older women outnumbering older men by a

**Funding:** This study was supported by Grants-in-aid for Scientific Research from the Ministry of Education, Science, Sports and Culture of Japan (18890055 and 20249041) (https://www.jsps.go.jp/english/e-grants/) and by the Mitsui Sumitomo Insurance Welfare Foundation (https://www.ms-ins.com/welfare/english/index.htm). The funders had no role in study design, data collection and analysis, decision to publish, or preparation of the manuscript.

**Competing interests:** The authors have declared that no competing interests exist.

factor of 1.3 overall and the gender imbalance increasing at the oldest age ranges [5]. It is expected that the proportion of the older adult population will continue to increase through the year 2045 [5]. There are currently 6.45 million older adults in Japan receiving long-term care due to functional impairments, and 70% of those receiving this type of care are women [6]. Many of these impairments are due to health conditions that are linked with MetS, including 15.2% of cases of impairment among older women caused by cardiovascular disease, and 15.4% caused by dementia [5].

Cardiovascular disease accounts for 18% of the mortality in Japanese women in their 60s, and increases to 24% in women in their 70s and 36% in women in their 80s [7]. For the aging population in particular, MetS may be a better indicator of cardiovascular risk than factors such as obesity alone. Although general measures of body mass, such as body mass index (BMI), are reliably related to cardiovascular mortality among adults in general, these relationships become more complicated during older adulthood [8]. When considering a relationship that has been termed the "obesity paradox," many studies have found that higher body mass, which is related to poorer cardiovascular outcomes in younger and middle adulthood, is often associated with a lower risk of these outcomes in older adults [9]. It is likely that changes in the general pattern of normal body composition during older adulthood are related to this paradoxical effect [10]. Individuals tend to gradually lose muscle mass in older adulthood, which can contribute to problems associated with frailty or sarcopenia [11,12]. When accompanied with increasing fat mass, particularly abdominal visceral fat, this tendency can contribute to sarcopenic obesity [13]. As a result, measures such as BMI may become less accurate with age, as the balance between muscle mass and fat mass shifts to proportionally less muscle and more fat [14].

Muscle mass content has been linked to cardiovascular health and mortality in several studies. Large epidemiological studies have found that, among older adults, radiographically assessed muscle mass is prospectively associated with lower all-cause mortality [8] as well as mortality due to cardiac-specific causes [15]. Studies based on cross-sectional data have also reported that older adults in the lowest quartile for muscle mass have elevated cardiovascular risk evaluations [16]. This relationship has also been identified more broadly in laboratory findings, as physiological research has found older adults with higher muscle mass exhibit better cardiac functioning during exercise [17]. However, there has been little previous research in this area that has included high-quality controls for body fat mass.

This study aimed to examine the relationship between body composition, assessed using accurate radiographic methods, and cardiovascular risk factors among women in older adulthood. This is the first study of which we are aware to use radiographic methods in this population. We hypothesized that within this population, greater muscle mass would be associated with lower risk of metabolic syndrome, after controlling for visceral fat mass.

## Materials and methods

Patients recruited for this study were healthy (defined as having no functional limitations in terms of their activities of daily living) older women receiving treatment at a single community clinic located in Tokyo, Japan. A consecutive series of female patients aged 65 years and older and receiving routine medical check-ups were screened for participation from September 1, 2005 to November 30, 2005. Exclusion criteria included factors potentially influencing visceral fat and skeletal muscle composition: a history of cancer, gastrointestinal tract surgery, or chronic viral hepatitis; subjects under treatment for endocrine diseases, diabetes, or heart failure; subjects on alpha-blocker, beta-blocker, beta-stimulator or hormone therapy (including glucocorticoids); and non-healthy subjects with serum albumin ≤3.0 g/dL, serum creatinine

>1.5 mg/dL, or blood hemoglobin ≤10.0 g/dL, as they might have been affected by such factors as abnormal fat metabolism and insulin resistance. A total of 158 patients were screened, with 90 satisfying the inclusion and exclusion criteria. The final sample had a mean age of 75.0 years (standard deviation [SD], 6.2 years). All assessments were conducted in a single session in the clinic at the time of treatment. The study protocol was approved by the Ethics Committee of the Research Institute of Aging Science (20050001), and all patients provided written informed consent prior to participation.

## Measurements

**Cardiovascular risk.** Clinical information regarding metabolic risk was collected as part of each patient's medical check-up. Blood sampling and height, weight, and waist circumference measurements were performed in the morning following a 12-hour overnight fast. Blood pressure was measured in the sitting position. BMI ($kg/m^2$) was calculated as weight in kilograms divided by the square of the height in meters. Serum triglyceride concentrations were measured enzymatically, and serum high-density lipoprotein (HDL) cholesterol concentration was measured using the heparin-$Ca^{2+}Ni^{2+}$ precipitation method. Plasma glucose concentration was assayed using the glucose oxidase method, and the HbA1c level was measured using high-performance liquid chromatography. Homeostasis model assessment of insulin resistance (HOMA-IR) was calculated as the fasting insulin level (μIU/mL) × fasting plasma glucose level (mg/dL)/405 [18,19]. These assays were performed at a commercial laboratory (SRL, Inc., Tokyo, Japan). The intra-assay coefficients of variation for the measurements were <5%.

We applied the International Diabetes Federation (IDF) criteria for the diagnosis of MetS [20]. According to IDF criteria for women of Japanese ethnicity, MetS is diagnosed when a waist circumference is ≥80 cm and two or more of the following four criteria are met: 1) HDL cholesterol level <1.29 mmol/L; 2) triglycerides level ≥1.7 mmol/L and/or fibrate treatment; 3) systolic blood pressure ≥130 mmHg and/or diastolic blood pressure ≥85 mmHg and/or treatment with antihypertensive medication, and; 4) fasting plasma glucose ≥5.6 mmol/L. For this study, abdominal circumference was excluded as a factor in evaluating MetS as it is a measure of visceral fat, which was observed directly using computed tomography (CT). Cardiovascular risk was thus defined in this study as the sum of the counts of the remaining MetS risk factors, producing a scale from 0 (no risk factors present) to 4 (all four non-obesity MetS factors present).

**Skeletal muscle mass.** Dual-emission X-ray absorptiometry (DEXA; model DELPHI-W, software version 11.2; Hologic, Boston, USA) was used to evaluate skeletal muscle mass. DEXA is a radiometric technique that has been shown to be more accurate and precise than alternative modes of physical measurement [21]. Appendicular skeletal muscle (ASM: kg) was calculated as the sum of lean soft tissue (nonfat, nonbone) mass in the arms and legs, which primarily represents skeletal muscle mass in the extremities. Validity and reproducibility of body composition data have previously been reported [22,23]. Based on this measure, the skeletal muscle mass index (SMI) was computed, adjusting for height as follows: SMI = ASM / height$^2$: $kg/m^2$ [24]. Since fat mass and muscle mass are strongly related, it was determined that SMI, with body size correction only for muscle mass, should not be used. Therefore, ASM was used instead of SMI in the basic analysis, and BMI was used for correction in all multivariable analyses. However, SMI was used for bivariate analyses (e.g., correlation) in which it was not possible to correct for body size by other means.

**Visceral fat mass.** CT (X Vision Scanner, Toshiba Medical Systems, Tokyo, Japan) was used to measure visceral fat mass. A previous study reported that this technique provides a more precise alternative measure of waist circumference [25]. Measures concerning visceral

fat area (VFA) and subcutaneous fat area (SFA) were obtained from a cross-sectional image at the umbilical level in the supine position using CT, involving a methodology that has been reported previously [26]. VFA ($cm^2$) and SFA ($cm^2$) were calculated using commercially available software (Fat Scan, N2 System, Osaka, Japan).

### Statistical methodology

Descriptive statistics are reported as mean ± SD in the text unless otherwise stated. Pearson's simple correlation coefficients between SMI and age or BMI, VFA, and HOMA-IR were calculated. Multiple regression analysis was performed to determine the association between ASM and metabolic risk factors. Data were analyzed using Stat View software (version 5.0, SAS Institute, Cary, NC, USA). In this study, ASM was used in the multiple regression analysis rather than SMI, because the metabolic risk factors included in the multiple regression included body mass, which is also a component of SMI.

## Results

Patient clinical characteristics are summarized in Table 1. The mean SFA was found to be markedly larger than the mean VFA ($170.2 \pm 75.8 \ cm^2$ vs $90.6 \pm 46.0 \ cm^2$). ASM was $13.0 \pm 1.8$ kg, SMI was $5.72 \pm 0.65$ ($kg/m^2$) using DEXA. The most common MetS risk factor was high blood pressure, occurring in 77.8% of patients, with 41.1% of the total sample undergoing antihypertensive treatment. This is consistent with the population of elderly Japanese women in general, for whom the prevalence of hypertension in the year 2000 was greater than 75% [27]. Despite excluding patients with diabetes from the study, the prevalence of high blood glucose was 31.1%. Most patients reported having never been smokers (93.3%), and very few had a history of cerebral infarction (3.3%) or ischemic heart disease (2.2%). The most common types of medications used included antihypertensives (41.4%) and statins (25.5%).

Fig 1 illustrates the correlations between SMI and age (A), BMI (B), VFA (C), and HOMA-IR (D). Older age was associated with significantly lower SMI ($r = -0.25$, $P = .01$), whereas SMI was positively related to BMI ($r = 0.79$, $P < .001$), VFA ($r = 0.46$, $P < .001$) and HOMA-IR ($r = 0.49$, $P < .001$).

Fig 2 illustrates the difference in the relationship between VFA and ASM among patients with higher and lower cardiovascular risk (defined as those with 0 or 1 MetS risk factors vs. those with 2–4 MetS risk factors). As this figure illustrates, there was no association between VFA and MetS, but as VFA increased a relationship between a significant relationship emerged, with greater muscle mass observed in patients with fewer MetS symptoms.

Multiple regression results, presented in Table 2, showed a relationship between body composition factors and metabolic cardiovascular risk. First, preliminary analyses of the bivariate relationships between these variables (Models 1a – 1c) indicated that, prior to including any controls, VFA was significantly related to metabolic risk ($\beta = 0.285$, $P = .007$, $r^2 = 0.08$), but there was no significant relationship between metabolic risk and either ASM ($\beta = -0.076$, $P = .470$, $r^2 = 0.006$) or BMI ($\beta = 0.114$, $P = .170$, $r^2 = 0.02$). When ASM and VFA were included in the model simultaneously (Model 2), regression results showed a stronger positive relationship between VFA and metabolic risk ($\beta = 0.391$, $P < .001$) along with a significant negative relationship between ASM and metabolic risk ($\beta = -0.246$, $P < .05$). These findings indicated a suppressor effect between ASM and VFA, whereby the positive relationship between these two variables appeared to have obscured the countervailing relationships between each of these factors and metabolic risk separately. These results persisted after controlling for patient age (Model 3) as well as for age and BMI together (Model 4), with similar standardized coefficient

**Table 1. Descriptive statistics for the clinical characteristics of the patient sample (n = 90).**

| Clinical Characteristics | Mean ± SD or n (%) | Range |
|---|---|---|
| Age (years) | 75.0 ± 6.2 | [65–91] |
| Body mass index (kg/m$^2$) | 22.1 ± 3.3 | [15.9–30.5] |
| Waist circumference (cm) | 82.5 ± 11.1 | [54.0–107.0] |
| *Blood Assay Values* | | |
| Glucose (mmol/L) | 5.39 ± 0.64 | [4.33–7.72] |
| Insulin (pmol/L) | 51.7 ± 25.6 | [1.4–132.8] |
| HOMA-IR | 1.81 ± 1.12 | [0.05–6.73] |
| *Body Composition* | | |
| Visceral fat area (cm$^2$) | 90.6 ± 46.0 | [17.5–227.1] |
| Subcutaneous fat area (cm$^2$) | 170.2 ± 75.8 | [27.1–353.9] |
| Appendicular skeletal muscle (kg) | 13.0 ± 1.8 | [9.1–20.5] |
| Skeletal muscle mass index (kg/m$^2$) | 5.72 ± 0.65 | [4.41–8.20] |
| *Metabolic syndrome (MetS) components* | | |
| High blood pressure, n (%) | 70 (77.8) | - |
| High serum triglycerides, n (%) | 6 (6.7) | - |
| Low HDL-cholesterol, n (%) | 20 (22.2) | - |
| High blood glucose, n (%) | 28 (31.1) | - |
| Number of MetS components[†] | 1.39 ± 0.89 | [0–4] |
| MetS components ≥2, n (%) | 26 (28.9) | - |
| *Smoking status* | | |
| Current smoker, n (%) | 4 (4.4) | - |
| Former smoker, n (%) | 2 (2.2) | - |
| Never smoked, n (%) | 84 (93.3) | - |
| *Medical History* | | |
| Cerebral infarction, n (%) | 3 (3.3) | - |
| Ischemic heart disease, n (%) | 2 (2.2) | - |
| *Current Medication* | | |
| Antihypertensive drugs, n (%) | 37 (41.1) | - |
| Fibrates, n (%) | 1 (1.1) | - |
| Statins, n (%) | 23 (25.5) | - |
| Vitamin D or Calcium, n (%) | 20 (22.2) | - |
| Bisphosphonates or Raloxifene, n (%) | 16 (17.7) | - |

HOMA-IR, homeostasis model assessment of insulin resistance.

estimates for ASM and VFA and negligible change in the model $r^2$ statistics between these successive models (Table 2).

## Discussion

The results of these analyses support the hypothesis that greater muscle mass is related to lower MetS risk in older Japanese women, after controlling for visceral fat mass. There was a strong and positive relationship between these two elements of body composition in this sample (older adult women with higher muscle mass also tended to have higher fat mass) and, as a consequence, the positive relationship between fat mass and cardiovascular risk obscured the negative relationship between muscle mass and cardiovascular risk when these factors were examined in isolation. In contrast, BMI was not significantly related to other elements of MetS in this sample, either alone or in combination with VFA and ASM. Additionally, the

**A**

**B**

**C**

**D**

**Fig 1. The correlation between SMI (corrected for body size) and age (A) or BMI (B) or VFA (C) or HOMA-IR (D).**

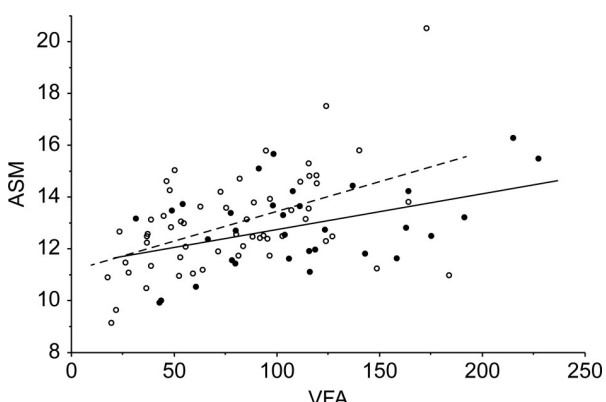

**Fig 2. The correlation between VFA and ASM (not corrected for body size) among patients with high and low cardiovascular risk.** Black spots and solid lines represent patients with two or more metabolic risk factors. White spots and broken lines represent patients with one or fewer risk factors.

**Table 2. Multiple regression analysis of the number of metabolic cardiovascular risk factors.**

| Variable | Model 1a - 1c[a] | Model 2[b] | Model 3[c] | Model 4[c] |
|---|---|---|---|---|
| $r^2$ | - | 0.130 | 0.131 | 0.132 |
| ASM | - 0.076 | - 0.246* | -0.257* | -0.276* |
| VFA | 0.285** | 0.391*** | 0.394*** | 0.368* |
| BMI | 0.114 | - | - | 0.049 |

*** P < 0.001,

** P < 0.01,

* P < 0.05

ASM, appendicular skeletal muscle; BMI, body mass index; VFA, visceral fat area.

Parameter estimates are presented in the form of standardized coefficients.

[a]Bivariate relationships between individual predictors and risk factors.

[b]Multiple regression with no control covariates.

[c]Includes control covariate adjusting for age.

interrelationship between VFA and ASM was found to differ somewhat between higher- and lower-risk groups, suggesting that the balance between fat and muscle content also has implications for the individual effect of these factors on cardiovascular risk. This relationship is illustrated by Fig 2, which shows how the relationship between VFA and ASM differs between groups of patients with lower and higher cardiovascular risk. This relationship appears to be different in a geriatric population than in adults at younger ages; even at the same weight, body composition in terms of fat and muscle mass differs for older adults. This suggests that among geriatric patients it is likely that the balance between fat and muscle mass, rather than the absolute mass of either, may be most important for positive health outcomes.

These findings broadly concur with those of a previous cross-sectional study of Korean older adults that found low muscle mass was linked to higher levels of cardiovascular risk [14]. However, that study included controls only for the BMI category, rather than using the type of direct measure of visceral fat used in this study.

Our findings emphasize the importance of accounting for the changes in body composition that occur during older adulthood. Increasing fat mass and decreasing muscle mass are changes in the normal aging process, making normal body composition in late life different from that of individuals in younger and middle adulthood [12]. Consequently, measures such as BMI, which do not differentiate between muscle and fat content, may not be as useful for older adults [14]. Previous research indicates that obesity is not reliably related to mortality risk in older adults [8], whereas it is a strong predictor of this risk in earlier stages of adulthood [9]. Shrinking muscle mass can act to reduce total body mass at the same time as body fat mass is increasing, a condition described as sarcopenic obesity [13].

A common shortcoming of research in the area of sarcopenic obesity is reliance on measures of body composition such as BMI, waist circumference, and whole-body fat that are prone to error and may not be appropriate for an older adult population [14]. One function of this study is to highlight the existence of more accurate and appropriate measures of body composition, including the use of radiographic techniques to directly assess the key measures of skeletal muscle mass and visceral fat mass. In contrast to the less direct and accurate BMI measure, these direct measurement techniques demonstrated clear relationships with cardiovascular risk factors. These results are also useful in providing some potential context for understanding the obesity paradox, whereby obesity appears to act as a protective factor against cardiovascular risk in some groups of older adults. Measures of obesity designed to

assess individuals with body composition patterns of early and middle adulthood may significantly mischaracterize obesity in older adulthood. The direct measure of visceral obesity used here demonstrated strong potential for use in this context, indicating a direct relationship between the body composition in older adulthood and cardiovascular risk, in contrast to BMI.

The clinical implications of this study relate to the independent relationships observed between both visceral fat and muscle mass and MetS. One interesting possibility suggested by these results is that cardiovascular risk may be less likely to increase among older women with visceral fat accumulation if muscle mass can be maintained in later life. Interventions to maintain and build muscle mass may be useful in achieving the goal of reducing negative cardiovascular outcomes in both obese and non-obese older adults. It also appears that reducing obesity may be beneficial for cardiovascular health even in later life, notwithstanding concerns regarding the obesity paradox, as long as obesity is assessed using age-appropriate tools. Clinicians treating older adults should aim to use more accurate and direct measures of visceral fat content, rather than relying on tools such as BMI or waist circumference that do not necessarily take the dynamics of body composition in older adulthood into consideration. Future research should aim to further delineate how body composition components at differing life stages may interact in varying degrees to influence cardiovascular risk, especially in older adulthood.

Limitations of this study include those inherent to the observational design. It is not possible to determine causality or the directionality of the relationships observed. For example, it is possible that individuals who have had long-term elevated cardiovascular risk also experience more rapid loss of muscle mass in later life. Additionally, the sample in this study was limited to older Japanese women without functional impairments. Further research is needed to examine the extent to which these results can be extended to men, to people of other ethnic backgrounds, and to those with a greater number of health issues. Furthermore, the relatively low r-squared statistics for the regression analyses suggest that factors other than those measured in this study may explain a larger proportion of the variability in cardiovascular risk in this population; those factors need to be more fully elucidated before the relationship between muscle mass and cardiovascular risk can be estimated accurately.

Nevertheless, these results have some important potential methodological and clinical implications. From a research standpoint, they demonstrate the importance of using accurate and direct measures of body composition when assessing the effects of risk factors, including muscle and fat content, on health outcomes in older adults. From a clinical perspective, they suggest that it is important to consider muscle mass and visceral fat mass as two distinct factors that may influence cardiovascular health in older adulthood. It is important to understand the key role that low muscle mass may play in increasing cardiovascular risk among older Japanese women, and the challenges in undertaking accurate assessment in relation to this key measure that changing patterns of body composition in later life may cause. As the Japanese population continues to age, it is critically important to understand factors that contribute to functional limitations among older adults so that changes in population health can be anticipated and mitigated.

## Acknowledgments

Editorial support, in the form of medical writing, assembling tables and creating high-resolution images based on authors' detailed directions, collating author comments, copyediting, fact checking, and referencing, was provided by Editage, Cactus Communications and supported by Nomura Clinic.

## Author Contributions

**Conceptualization:** Kazushi Nomura, Yasuyoshi Ouchi, Masahiro Akishita.

**Data curation:** Kazushi Nomura, Taro Kojima, Tetsuro Nakamura, Atsushi Araki.

**Formal analysis:** Kazushi Nomura, Masato Eto, Masahiro Akishita.

**Funding acquisition:** Kazushi Nomura, Yasuyoshi Ouchi, Masahiro Akishita.

**Investigation:** Kazushi Nomura, Sumito Ogawa, Taro Kojima, Katsuya Iijima, Masahiro Akishita.

**Methodology:** Kazushi Nomura, Masato Eto, Masahiro Akishita.

**Project administration:** Kazushi Nomura, Masato Eto, Masahiro Akishita.

**Resources:** Kazushi Nomura, Yasuyoshi Ouchi, Masahiro Akishita.

**Software:** Kazushi Nomura, Atsushi Araki.

**Supervision:** Masahiro Akishita.

**Validation:** Kazushi Nomura, Masato Eto, Sumito Ogawa, Taro Kojima, Katsuya Iijima, Masahiro Akishita.

**Visualization:** Kazushi Nomura.

**Writing – original draft:** Kazushi Nomura, Masato Eto.

**Writing – review & editing:** Kazushi Nomura, Sumito Ogawa.

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
