## [Decision Letter · Decision Letter 0]

18 Nov 2020

Association between low muscle mass and metabolic syndrome in elderly Japanese women

PONE-D-20-28475

Dear Dr. Nomura,

We’re pleased to inform you that your manuscript has been judged scientifically suitable for publication and will be formally accepted for publication once it meets all outstanding technical requirements.

Kind regards,

Tatsuo Shimosawa, M.D., Ph.D.

Academic Editor

PLOS ONE

1.PLOS requires an ORCID iD for the corresponding author in Editorial Manager on papers submitted after December 6th, 2016. Please ensure that you have an ORCID iD and that it is validated in Editorial Manager. To do this, go to ‘Update my Information’ (in the upper left-hand corner of the main menu), and click on the Fetch/Validate link next to the ORCID field. This will take you to the ORCID site and allow you to create a new iD or authenticate a pre-existing iD in Editorial Manager. Please see the following video for instructions on linking an ORCID iD to your Editorial Manager account: https://www.youtube.com/watch?v=_xcclfuvtxQ

<h1>** **</h1>

Reviewers' comments:

Reviewer's Responses to Questions

**Comments to the Author**

1. Is the manuscript technically sound, and do the data support the conclusions?

Reviewer #1: Yes

2. Has the statistical analysis been performed appropriately and rigorously? 

Reviewer #1: Yes

3. Have the authors made all data underlying the findings in their manuscript fully available?

Reviewer #1: Yes

4. Is the manuscript presented in an intelligible fashion and written in standard English?

Reviewer #1: Yes

5. Review Comments to the Author

Reviewer #1: This is a cross-sectional study investigated the relationship among skeletal muscle mass, visceral fat mass and cardiovascular risk factors among women in older adulthood.

They showed that greater muscle mass is related to lower cardiovascular risk in older adult women, after controlling for visceral fat mass. As there was a positive relationship between skeletal muscle mass and visceral fat mass, the positive relationship between fat mass and cardiovascular risk obscured the negative relationship between muscle mass and cardiovascular risk when these factors were examined in isolation.

The authors responded to the previous review and revised the manuscript properly.

I think this article is very well written and worth for publication.

6. PLOS authors have the option to publish the peer review history of their article (what does this mean?). If published, this will include your full peer review and any attached files.

Reviewer #1: No

---

## [Editor Report · Acceptance letter]

23 Nov 2020

PONE-D-20-28475 

Association between low muscle mass and metabolic syndrome in elderly Japanese women 

Dear Dr. Nomura:

I'm pleased to inform you that your manuscript has been deemed suitable for publication in PLOS ONE. Congratulations! Your manuscript is now with our production department. 

Kind regards, 

on behalf of

Prof. Tatsuo Shimosawa 

Academic Editor

PLOS ONE